

# Comparative genomics of bacteria from amphibian skin associated with inhibition of an amphibian fungal pathogen, *Batrachochytrium dendrobatidis*

Noah Wax[1], Jenifer B. Walke[1,2], David C. Haak[3] and Lisa K. Belden[1]

[1] Department of Biological Sciences, Virginia Polytechnic Institute and State University (Virginia Tech), Blacksburg, VA, United States of America
[2] Department of Biology, Eastern Washington University, Cheney, WA, United States of America
[3] School of Plant and Environmental Sciences, Virginia Polytechnic Institute and State University (Virginia Tech), Blacksburg, VA, United States of America

Corresponding author
Noah Wax, noahw15@vt.edu

## ABSTRACT

Chytridiomycosis, caused by *Batrachochytrium dendrobatidis* (Bd), is a skin disease associated with worldwide amphibian declines. Symbiotic microbes living on amphibian skin interact with Bd and may alter infection outcomes. We completed whole genome sequencing of 40 bacterial isolates cultured from the skin of four amphibian species in the Eastern US. Each isolate was tested *in vitro* for the ability to inhibit Bd growth. The aim of this study was to identify genomic differences among the isolates and generate hypotheses about the genomic underpinnings of Bd growth inhibition. We identified sixty-five gene families that were present in all 40 isolates. Screening for common biosynthetic gene clusters revealed that this set of isolates contained a wide variety of clusters; the two most abundant clusters with potential antifungal activity were siderophores (N=17 isolates) and Type III polyketide synthases (N=22 isolates). We then examined various subsets of the 22 isolates in the phylum Proteobacteria for genes encoding specific compounds that may inhibit fungal growth, including chitinase and violacein. We identified differences in *Agrobacterium* and *Sphingomonas* isolates in the chitinase genes that showed some association with anti-Bd activity, as well as variation in the violacein genes in the *Janthinobacterium* isolates. Using a comparative genomics approach, we generated several testable hypotheses about differences among bacterial isolates from amphibian skin communities that could contribute to variation in the ability to inhibit Bd growth. Further work is necessary to explore and uncover the various mechanisms utilized by amphibian skin bacterial isolates to inhibit Bd.

## INTRODUCTION

Vertebrates are host to a wide range of microorganisms that form communities inhabiting different body regions, including the gut, mouth, and skin (*Ogunrinola et al., 2020*). These

symbiotic microbial communities play a role in host health; however, many of the specific mechanisms that mediate these responses are unknown (*Cénit et al., 2014*). In many hosts, these microbial communities appear to provide a critical first line of defense against pathogen entry (*Swaney & Kalan, 2021*). For example, in humans, some skin bacteria may prevent infection by *Staphylococcus aureus* by producing antimicrobial peptides that specifically target *S. aureus* (*Nakatsuji et al., 2017*). Similarly, in amphibians, skin bacterial communities may play a role in preventing or limiting infection by the potentially-lethal fungal pathogen *Batrachochytrium dendrobatidis* (Bd) by producing anti-fungal compounds or outcompeting Bd for resources (*Brucker et al., 2008a*; *Brucker et al., 2008b*).

*Batrachochytrium dendrobatidis*, and the more recently identified *B. salamandivorans*, are the causative agents of the disease chytridiomycosis, which has contributed to large-scale global declines of amphibian populations (*Berger et al., 1998*; *Martel et al., 2013*; *Scheele et al., 2019*) The composition of bacterial communities found on amphibian skin can sometimes predict host responses following exposure to Bd (*Becker et al., 2015a*; *Walke et al., 2015a*). These skin bacterial communities, as well as the gene networks within them, can also be altered by the presence of Bd (*Rebollar et al., 2018*). One example of a bacterium that produces anti-fungal compounds is *Janthinobacterium lividum*, which has been isolated from multiple amphibian species and produces indole-3-carboxaldehyde, as well as violacein. These compounds can inhibit Bd growth *in vitro,* and treatment of frogs with *J. lividum* in lab experiments can reduce morbidity and mortality (*Brucker et al., 2008b*; *Harris et al., 2009*) . While many bacteria collected from amphibian skin have anti-fungal properties against Bd *in vitro* (*Woodhams et al., 2015*), less is known about the diversity of anti-fungal metabolic pathways in these communities of amphibian skin bacteria, or whether common genes tend to contribute to Bd inhibition across these bacterial taxa.

Some recent studies have used genomic approaches to try to understand the species interactions in amphibian skin microbial communities. For instance, *Rebollar et al. (2018)* used shotgun metagenome sequencing of DNA samples from skin swabs of frogs from Bd-endemic and Bd-naïve sites. They found gene classes involved in biosynthesis of secondary metabolites, cellular communication and membrane transport enriched in the samples from the Bd-endemic site. They also found genes involved in the production of anti-fungal compounds, such as prodigiosin, indole-3-carboxaldehyde, and 2,4-diacetylphloroglucinol in both the Bd-endemic and Bd-naïve sites (*Rebollar et al., 2018*). Interestingly, these genes were found in different genera depending on the site, which indicates that there are multiple bacterial species capable of producing compounds that may inhibit Bd growth.

Advances in whole genome sequencing have made it easier to obtain genetic information from a wide variety of organisms. Prokaryote genomes are of interest due to the wide variety of bioactive secondary metabolites they produce, such as hedamycin, candicidin and avermectin, which can be used in many applications (*Lautru et al., 2005*; *Franke, Ishida & Hertweck, 2012*; *Risdian, Mozef & Wink, 2019*). In prokaryotes, genes related to the synthesis of secondary metabolites are often located near each other in regions described as biosynthetic gene clusters (BGC) (*Blin et al., 2019*). These gene clusters have been linked to the production of antibiotics, siderophores, and other medically relevant

drugs (*Newman & Cragg, 2012*). BGCs that may be important for bacterial anti-fungal activity include Type III polyketide synthases and siderophores (*Risdian, Mozef & Wink, 2019*). Recent developments of bioinformatic tools are allowing deeper investigations of microbial communities and narrowing the search for these gene clusters, and for other genes that encode novel bioactive secondary metabolites. For example, the development of tools that do not require a genome assembly, so called "assembly free" approaches, such as antiSMASH (*Blin et al., 2019*), have allowed for the prediction of biosynthetic gene clusters directly from genomic sequence data, thus, reducing computational time. Assembly based programs have also seen dramatic improvement in algorithms, reducing assembly times. Together, these advances are facilitating discoveries in natural systems that were previously considered too challenging.

In previous work, we collected 719 bacterial isolates from four species of Virginia amphibians and tested their ability to inhibit Bd *in vitro* (*Walke et al., 2015b*; *Walke et al., 2017*). Based on their ability to inhibit the growth of Bd, 47 of these isolates, spanning 13 families and 15 genera, were selected for whole genome sequencing. The aim of this study was to use comparative genomics approaches to identify genes and gene clusters that were associated with variation in Bd growth inhibition among sets of related isolates.

## METHODS

### Isolate collection

Bacteria used in this study were originally isolated from the skin of four species of Virginia amphibians as described in *Walke et al. (2015b)*: American bullfrogs (*Lithobates catesbeianus*), spring peepers (*Pseudacris crucifer*), American toads(*Anaxyrus americanus*), and Eastern newts(*Notophthalmus viridescens*). These four species are all common amphibians native to the Eastern US. They all breed in ponds and wetlands during the spring and summer and have an aquatic larval stage that metamorphoses into a more terrestrial juvenile or adult.

A total of 719 bacterial isolates were collected per methods in *Walke et al. (2015b)*. Amphibian handling protocols were approved by the Virginia Tech Institutional Animal Care and Use Committee (IACUC protocols 08-042-BIOL, 10-029-BIOL and 12-040-BIOL). Briefly, all animals were rinsed with sterile water and then swabbed with two sterile rayon swabs. The first swab was used to assess the whole bacterial community *via* 16S rRNA gene amplicon sequencing. The other swab was plated on R2A media to assess the culturable bacterial community. After incubating at room temperature for 14 days, individual colonies were isolated from plates based on margin, form, whole colony color, elevation and substance. These were serially plated until pure isolates were obtained and initially identified *via* Sanger sequencing of the 16S rRNA gene (primers 8F and 1492R). Cultured isolates were then tested for the ability to inhibit Bd growth *in vitro*, using a quantitative 96-well plate assay (*Bell et al., 2013*; *Becker et al., 2015b*). Optical density values were used to score isolates on a range of completely inhibitory (100) to those with no effect on Bd growth (0), and even a few with negative inhibition, indicative of facilitation of Bd growth (<0). Results comparing the full set of culturable and amplicon-based skin

communities were previously published in *Walke et al. (2015b)*, and results examining the bacterial skin community in relation to Bd growth inhibition were published in *Walke et al. (2017)*.

For the present study, a subset of 47 of the cultured isolates in 15 genera were selected for whole genome sequencing from the complete set of 719 isolates. Our approach was to select sets of related isolates based on 16S rRNA gene sequencing that varied in their ability to inhibit Bd, so that we could identify genomic differences that could potentially contribute to variation in Bd inhibition.

### DNA extraction

DNA from 20 of the 47 isolates sent for sequencing had been previously extracted (*Walke et al., 2015b*). The other 27 isolates used in this study needed to be cultured from freezer stocks and have their DNA re-extracted. For these, prior to DNA extraction, isolates were cultured at room temperature in 750 µl of 1% tryptone broth, shaking at 150 rpm for 24 h. After 24 h, we centrifuged the cultures at 7,500 RPM for 10 min. The resulting pellet was resuspended in 180 µl of lysis buffer containing 20mg/ml lysozyme. DNA extraction was completed using a DNeasy blood and tissue kit (Qiagen Inc., Valencia, CA, USA) following the gram-positive bacteria protocol provided by the manufacturer. DNA was eluted in 150 µl of sterile Milli-Q water.

### Sequencing and genome assembly

Extracted DNA was sent to the Duke Center for Genomic and Computational Biology for library construction and sequencing on the Illumina Hi-Seq 4000 platform, 2 × 150 bp. Samples were sequenced on a single lane. This generated an average of 2.2 Gbp of reads per sample library, with an average Q score of 38, and 90% of reads >Q30 (Table S1). Raw reads were adapter trimmed using Trimmomatic *v.* 0.35 (*Bolger, Lohse & Usadel, 2014*) with default settings including standard Illumina adapters, and were visually checked for quality using fastqc (*Andrews, 2010*). Processed raw reads were *de novo* assembled using Minia *v.* 2.0 (*Chikhi & Rizk, 2013*) with the command line arguments, -kmer 131 -abundance -min 3. The kmer choice was based on comparative assembly of a subset (3) of data for kmer lengths 21–141 with a step of 10. For the average assembly, the total length was 5.2 Mbp (range 2.8:8.9 Mbp), with 45 total contigs (range 11:211). The average N50 was 357 kbp (range 73:1, 105 kbp). The average L50 was 6.9 (2:22), and GC content was 55% (range 34:72%) (Table S2). Minia generated contig files were used for downstream analyses.

### Annotation

To verify the identities of the isolates prior to annotation, we used BBTools *v.* 38.86 (*Bushnell, 2018*) to obtain average nucleotide identity scores between individual isolates and their most closely related bacteria. Of the 47 genome assemblies, 40 unambiguously matched the initial isolate 16S identity and were used for subsequent analyses (Table S3, which includes isolate genera, Bd inhibition scores, and the amphibian host). To assess assembly completeness(accuracy of assembled orthologs), we analyzed the genome assemblies for Benchmarking Universal Single-Copy Orthologs (BUSCOs), which are genes that are expected to be present in closely related bacteria. To do this, we used

BUSCO *v.* 4.1.3 (*Simão et al., 2015*) with *v.* 10 BUSCO databases. BUSCO does not have databases for every genus, so assemblies were evaluated with the lowest taxonomic database available for that isolate. Nine assemblies were evaluated using class level databases, 29 were evaluated using order level databases and two were evaluated using genus level databases. All genome assemblies contained at least 97% of expected BUSCOs, suggesting the genomes were complete, and thus usable for subsequent analyses. To generate the General Feature Format (GFF) files necessary for analyses with PIRATE *v.* 1.0.3 (*Bayliss et al., 2019*), the 40 genomes were annotated using Prokka *v.* 1.14.6 (*Seemann, 2014*). To find orthologous groups and to functionally annotate the genomes, the FASTA files containing the amino acid sequences of the predicted coding sequences were annotated using eggNOG-mapper *v.* 2.0 with eggNOG database *v.* 5.0 (*Huerta-Cepas et al., 2017*; *Huerta-Cepas et al., 2019*; *Buchfink, Xie & Huson, 2015*).

## Comparative genomics

Our general approach was to identify unique genes and metabolic pathways, and differences in those shared genes and pathways that may contribute to variation in the ability of isolates to inhibit Bd growth. Due to the taxonomic diversity of the isolates, this was done at various scales. We initially created a whole genome tree and labelled the tips with the inhibition values. This tree allowed us to visualize the relationships among all the isolates and to determine that there was no clear phylogenetic pattern of Bd inhibition in our dataset. PIRATE was used to create and align the core genome of our isolates. The tree was generated using this core genome alignment and RAxML HPC version 8.2.12 (*Stamatakis, 2014*) (Fig. 1). RAxML was run using a random number seed for parsimony inferences, rapid bootstrapping with 1,000 replicates and the GTRCAT nucleotide substitution model. Then we compared subsets of the isolates based on phylum, class, and genus. Shared and unique gene families were identified, and we then looked for differences in shared biosynthetic gene clusters. In addition, for subsets of genera, we looked at genes for specific compounds, such as chitinase, that play a role in anti-fungal activity (*Brucker et al., 2008a*; *Brucker et al., 2008b*; *Le & Yang, 2019*).

Gene families are sets of genes that originated through duplication of an ancestral gene and have similar functions. To identify both shared and unique gene families among all 40 isolates, the GFF files produced by Prokka were run through PIRATE. Based on user defined amino acid sequence similarity thresholds, PIRATE identifies orthologous gene families in bacterial genomes. This was also done with subsets of the genomes that were divided based on phylum, class, order, and genus. We used core genome alignments from these subsets to try to identify patterns related to Bd inhibition. PIRATE was run on subsets consisting of the Proteobacteria ($N = 22$, and further division of classes Alphaproteobacteria ($N = 7$), Betaproteobacteria ($N = 7$) and Gammaproteobacteria ($N = 8$)), Bacteroidetes ($N = 8$), Actinobacteria ($N = 6$) and Firmicutes ($N = 4$). The 22 Proteobacteria were further divided by order and genus for analysis. The Bacteroidetes and Actinobacteria phyla were divided by genus because the isolates within them only differed at the genus level. To detect gene families present at lower taxonomic levels, PIRATE was run on the Burkholderiales order and a total of eight genera.

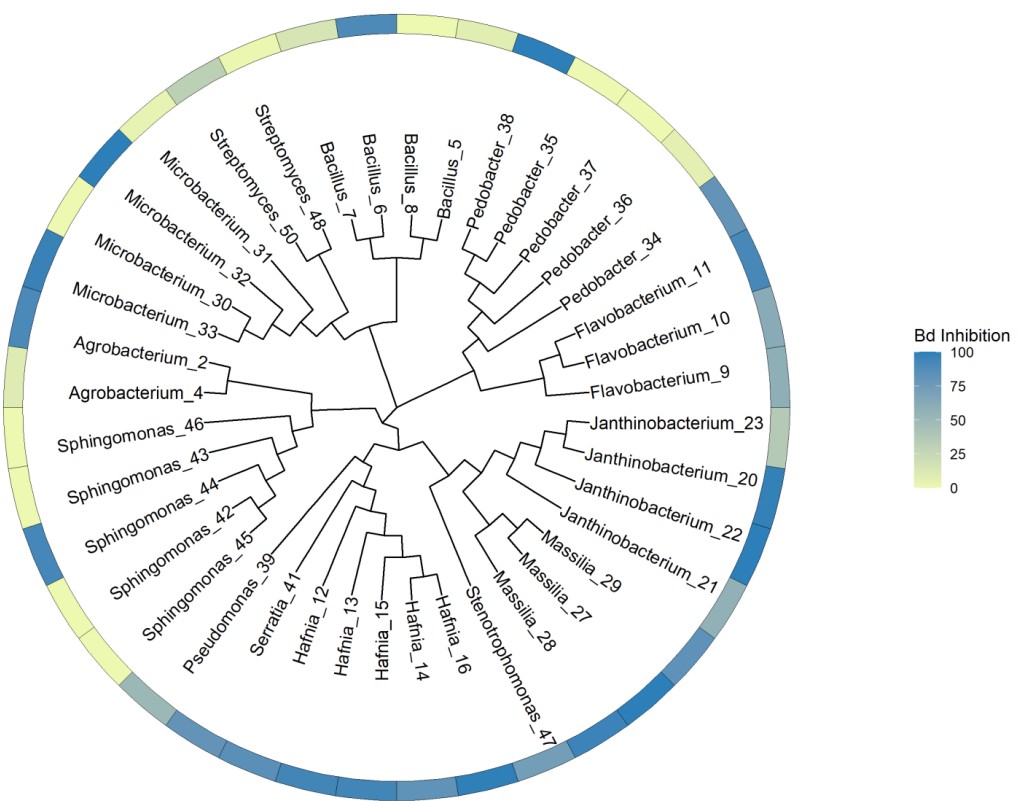

**Figure 1** **Phylogenetic tree constructed from whole genome sequences of 40 bacteria isolated from amphibian skin.** Bd inhibition values were obtained from *in vitro* challenge assays (*Walke et al., 2017*). The inhibition values refer to the mean inhibition. 100 = complete Bd inhibition (blue), 0 = no inhibition (yellow). For the purposes of this figure, isolates with negative scores indicating potential facilitation of growth ($N = 7$) were set at zero. Isolates are labeled with genus and sequencing ID number (Table S3).

To identify predicted biosynthetic gene clusters potentially associated with anti-fungal activity, all 40 genome assemblies, as well as their respective GFF files, were run through antiSMASH version 5.0 (*Blin et al., 2019*) using the strictest settings. We identified 28 different biosynthetic gene clusters in our 40 isolates. Terpene synthesis clusters were the most common and were identified in 33 of our isolates, followed by Type III polyketide synthases in 22 isolates and siderophores in 17 isolates. The structural and functional diversity of terpenes makes them difficult to compare across genera, so we compared genes within terpene synthesis clusters only within genera, as described below. We focused more on the Type III polyketide synthases and the siderophores, which can have anti-fungal activity (*Sulochana et al., 2014*; *Risdian, Mozef & Wink, 2019*).

To examine shared and unique genes in these two clusters, individual FASTA files containing nucleotide sequences of the predicted Type III polyketide synthase clusters and siderophore clusters were imported into Geneious Prime *v.* 2020.2.1 (Biomatters Ltd., Auckland, New Zealand). Genes that were shared among multiple isolates were then aligned using the Mafft plugin *v.* 1.4.0 (*Katoh et al., 2002*) in Geneious Prime. These alignments

were then concatenated and exported in phylip format and a maximum likelihood tree was generated using RAxML HPC *v.* 8.2.12 with the same parameters as previously mentioned. Bd inhibition values were then mapped to the tips of the trees that were generated to see if there were any observable patterns related to inhibition.

Only a single gene, iucC, was present in all predicted siderophore clusters across the various genera. Only one gene that encoded a Type III polyketide synthase was found in all Type III polyketide synthase clusters across genera. Because there were few genes shared at higher taxonomic levels, we then started looking for common genes in shared biosynthetic gene clusters within genera. Four genera were examined. *Bacillus* isolates ($N = 4$) shared a predicted terpene synthesis cluster, a predicted siderophore cluster and a predicted Type III polyketide synthase cluster. *Pedobacter* isolates ($N = 5$) shared a predicted siderophore, Type III polyketide synthase and a terpene synthesis cluster. *Microbacterium* isolates ($N = 4$) shared a Type III polyketide synthase and a terpene synthesis cluster. *Sphingomonas* isolates ($N = 5$) shared a Type III polyketide synthase cluster and a terpene synthesis cluster. For each of the genera we examined, shared genes within each BGC were aligned, RAxML (*Stamatakis, 2014*), was used to generate a phylogenetic tree using the same parameters mentioned previously, and Bd inhibition values were mapped to the tips of the resulting tree. By observing which tips had the highest Bd inhibition values we generated hypotheses about which genes to focus on for single nucleotide polymorphisms. We also used the patterns generated by mapping Bd inhibition values to these trees to generate hypotheses about potential involvement of various BGCs in Bd inhibition within specific genera.

To further explore potential links to anti-fungal activity, four genera were selected, *Bacillus* ($N = 4$; Bd inhibition: −8, 8, 15, 90), *Flavobacterium* ($N = 3$; Bd inhibition: 58, 62, 92), *Microbacterium* ($N = 4$; Bd inhibition: −21, 4, 97, 100) and *Sphingomonas* ($N = 5$; Bd inhibition: −30, −11, −9, −2 and 93). These genera were selected for further analysis because the isolates within them showed the greatest difference in their ability to inhibit Bd growth. For each of these four genera, the genomes of the 10 most closely related bacteria were downloaded from the National Center for Biotechnology Information (NCBI) and annotated using Prokka. These annotations were used for analysis in PIRATE to determine shared and unique genes families among the isolates within each genus. Only those proteins predicted by Prokka and eggNOG were retained for further analysis. Predicted proteins from the unique gene families identified from the isolates with greater Bd growth inhibition scores were characterized *via* BLASTp on the NCBI website.

Annotated genomes from individual isolates were then analyzed for genes that could potentially explain variation in Bd growth inhibition assays. Two isolates in the genus *Agrobacterium* (Bd inhibition: 10 and 91) contained a shared predicted chitinase gene. This gene was then translated, and the resulting amino acid sequences were aligned using BLASTp from the NCBI website. We also found a predicted chitinase in the only *Sphingomonas* isolate that was able to inhibit Bd; it was lacking from the other four isolates in that genus. Violacein is another compound that inhibits the growth of Bd (*Woodhams et al., 2018*). Genes that code for the enzymes necessary to produce the compound violacein were identified in the genomes of three out of the four *Janthinobacterium* isolates (Bd inhibition: 36, 98 and 101; Bd inhibition of isolate without genes for all enzymes: 57).
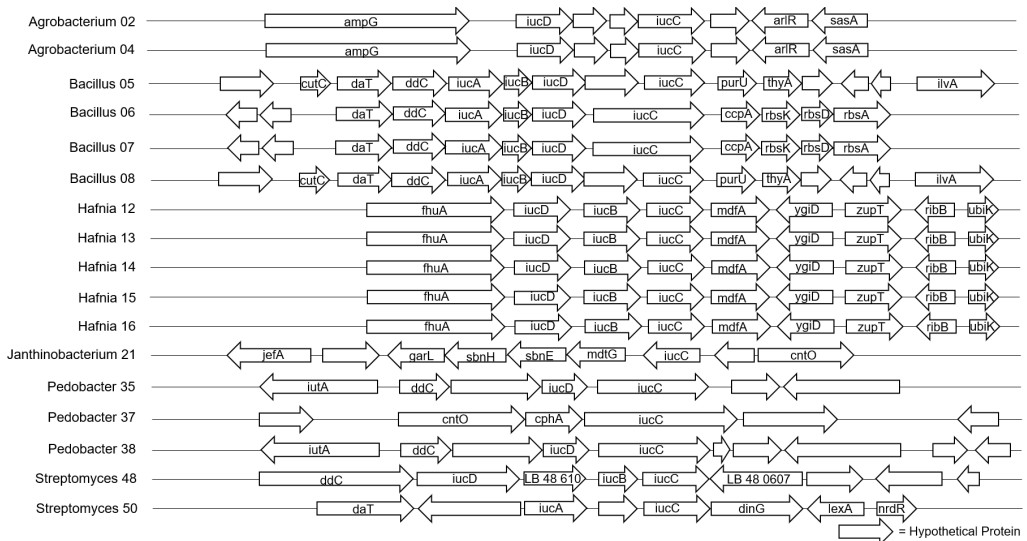

**Figure 2** **Alignment of predicted siderophore biosynthetic gene clusters found in bacterial isolates** **($N = 17$).** Only one gene, iucC was found in all 17 isolates that contained a siderophore BGC. Isolates are labeled with genus and sequencing ID number (Table S3).

These five genes (vioA, vioB, vioC, vioD and vioE) were translated and aligned using Clustal Omega (*Sievers et al., 2011*).

# RESULTS

PIRATE identified a total of 67,039 gene families across the 40 genomes analyzed. Only sixty-five (<1%) of those gene families were found in all isolates. In the Proteobacteria, there were 31,419 distinct gene families across the twenty-two isolates, in the Bacteroidetes, there were 14,561 gene families across the eight isolates, in the Actinobacteria, there were 15,267 gene families across the six isolates, and in the Firmicutes, there were 8,784 gene families across the four isolates.

The 40 isolates we examined contained over 26 distinct biosynthetic gene clusters. The most common biosynthetic gene cluster we found was for terpene synthesis, which was identified in 33 of the isolates (Table S1). However, this is a very large group of compounds and pathways, and there were zero shared genes found among our isolates, so we could not pursue further genomic analysis for these in the full dataset, and given the diversity of these compounds, we did not pursue them at other taxonomic levels either. Siderophore biosynthetic gene clusters were identified in 17 of the isolates (Fig. 2), and Type III polyketide synthase biosynthetic gene clusters were identified in 22 of the isolates (Fig. 3). We did not observe a clear pattern of correlation with Bd inhibition based on the *iucC* or Type III polyketide synthase gene trees. However, there were some shared genes in these pathways for several genera that we were able to analyze further.

The four *Bacillus* isolates shared a predicted siderophore cluster (six genes in common) and a Type III polyketide synthase cluster (seven genes in common) that we analyzed
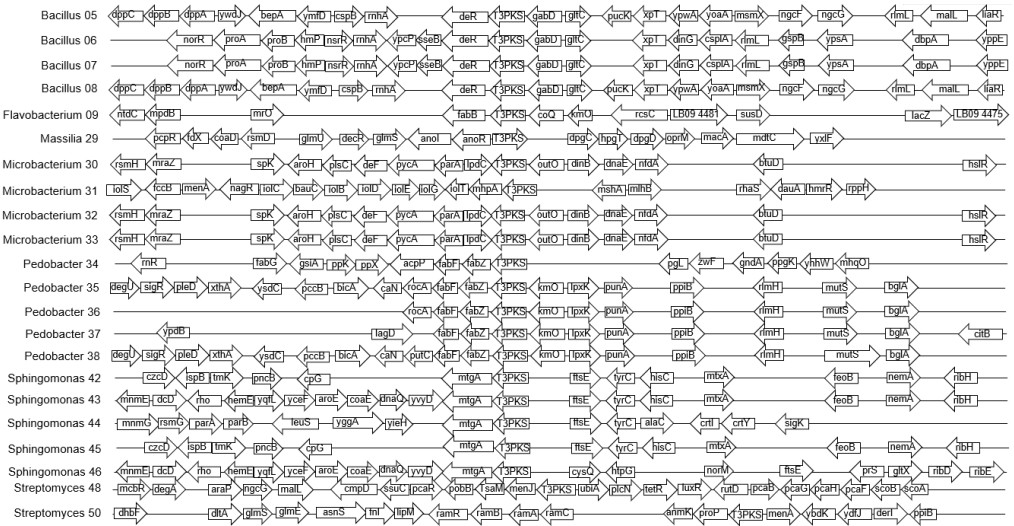

**Figure 3 Alignment of predicted Type III polyketide synthase biosynthetic gene clusters found in bacterial isolates (N = 22).** Hypothetical proteins not shown. Isolates are labeled with genus and sequencing ID number (Table S3).

further. The nucleotide sequences of the siderophore and Type III polyketide synthase genes were identical in two *Bacillus* isolates with very different Bd inhibition scores of 90 and 15. Three of five *Pedobacter* isolates had predicted siderophore gene clusters. Two of these isolates (no Bd inhibition, -2 ; strong Bd inhibition, 100) shared seven genes. This set of seven genes had an average nucleotide identity (ANI) of 86%. While there were differences in the nucleotide sequences of these shared genes, with only two isolates we could not look at their differences as related to inhibition. All five *Pedobacter* isolates shared seven genes in Type III polyketide synthase clusters. Based on the patterns seen after mapping Bd inhibition values to the tips of the phylogenetic tree generated from the gene alignment, the differences in these genes may contribute to variation in ability to inhibit Bd growth (Fig. S1). All four *Microbacterium* isolates shared two genes in a predicted Type III polyketide synthase cluster. In the five *Sphingomonas* isolates, we found predicted Type III polyketide synthase clusters with seven genes in common. While there were differences in these shared genes in the *Microbacterium* and *Sphingomonas* isolates, there were not clear correlations of these differences and Bd inhibition in our dataset (Figs. S2 and S3).

Chitinase was an additional anti-fungal compound we examined. Of the 40 isolates, there were two with chitinase genes in the genus *Agrobacterium* and five in the genus *Sphingomonas*. First, we examined the two *Agrobacterium*, which differed greatly in their ability to inhibit Bd growth (inhibition scores of 10 and 91). We found that these isolates shared a potential chitinase gene that was 639 base pairs (213 amino acids) long, and differed by four amino acids (Fig. 4). We identified the following substitutions between these two isolates: Val45Ala, Arg79Ile, Gln99Arg, and Arg195Lys. In addition to the differences seen in this shared predicted chitinase gene, the more inhibitory isolate contained a second predicted chitinase gene (897 bp) that was completely absent in the less inhibitory isolate.

**Figure 4** Amino acid alignment of a shared predicted chitinase between two *Agrobacterium* isolates, labeled *Agrobacterium* 04 (Bd inhibition = 10) and *Agrobacterium* 02 (Bd inhibition = 91), that differed in ability to inhibit Bd growth.

Of the five *Sphingomonas* isolates, only one was strongly Bd inhibitory (score = 93, others = $-30, -11, -9, -2$). The inhibitory *Sphingomonas* contained a predicted chitinase gene that was 657 base pairs (219 amino acids) long and was absent in the other four isolates.

We also examined the genes that contribute to the production of violacein in *Janthinobacterium* isolates. Three of the four *Janthinobacterium* isolates (inhibition scores = 37,98,101) contained all five genes necessary to produce violacein. The fourth *Janthinobacterium* isolate (inhibition score = 57) contained only the vioA gene (Fig. 5). The four amino acids $Arg^{64}$, $His^{163}$, $Lys^{269}$, and $Tyr^{309}$ are critical for the functionality of the protein vioA (*Füller et al., 2016*), and these amino acids were present in the vioA protein of all three of our *Janthinobacterium* isolates that had the complete violacein gene set. In the least inhibitory *Janthinobacterium* isolate of the three capable of violacein production (score =37), the vioA gene was 1,308 base pairs long, and after translation it was found that it differed from the two more inhibitory isolates by eight amino acids. There were also differences in the other violacein genes in this less inhibitory isolate. The vioB gene differed by twenty-four amino acids, the vioC gene by six amino acids, the vioD gene by thirteen amino acids, and the vioE gene by eight amino acids. $Tyr^{17}$, $Ser^{19}$, $Phe^{50}$, $Asn^{51}$, $Glu^{66}$, and $Arg^{172}$ are important for the vioE protein to catalyze reactions (*Hirano et al., 2008*). All of these amino acids except for $Glu^{66}$ were present in the vioE protein sequences in our isolates. The nucleotide differences in the vioA-E genes in three of the *Janthinobacterium* are highlighted by a mauve alignment (Fig. 6).

## DISCUSSION

The 40 isolates we studied contained over 26 distinct biosynthetic gene clusters, which is not surprising given the taxonomic breadth of our samples. Of the 26 gene clusters, we looked more closely at the three that were present in the most isolates and were potentially important for bacteria-fungi interactions: terpenes, Type III polyketide synthases and siderophores.

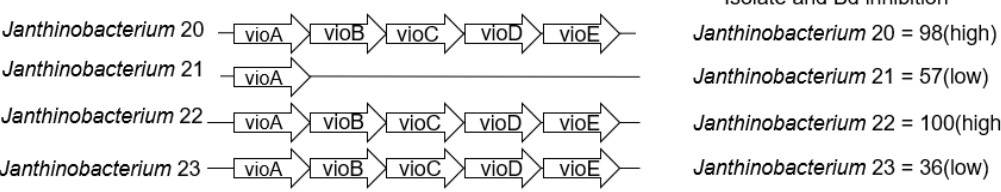

**Figure 5  Presence/absence of genes vioA–vioE necessary to produce violacein in the four *Janthinobacterium* isolates.**  Isolates are labeled with genus and sequencing ID number (Table S3).

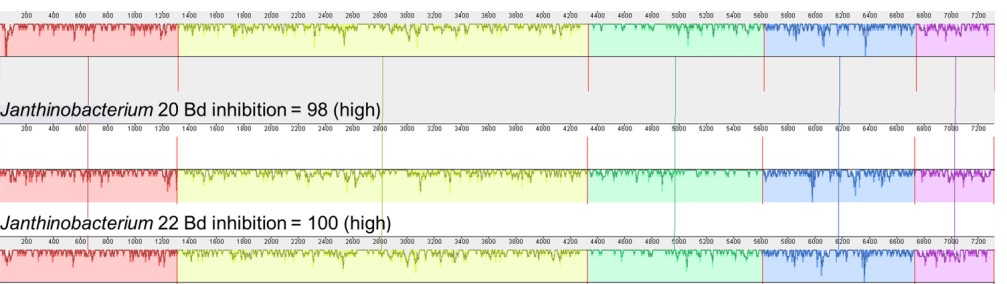

**Figure 6  Mauve alignment of violacein genes vioA–vioE found in the three *Janthinobacterium* isolates containing all five genes (Bd inhibition scores = 98, 100 and 36).**  Isolates are labeled with genus and sequencing ID number (Table S3).

Terpenes are a very broad class of molecules that are structurally and functionally diverse, and include antimicrobial compounds produced by plants, fungi and some bacteria, including the Actinomycetes and Streptomycetes (*Can Başer & Buchbauer, 2015*; *Citron et al., 2012*). A study of bacteria in the genera *Janthinobacterium* and *Duganella* found possible involvement of terpene production in fungal growth inhibition (*Haack et al., 2016*). While all terpenes have the same general chemical formula, their structural diversity makes it difficult to predict their function based solely from genetic information. We found that while there were many terpene gene clusters present in our dataset, there were very few shared genes among our isolates, such that we could not speculate on function as related to Bd inhibition for genes in this cluster.

Polyketides, sometimes referred to as polyketide natural products are secondary metabolites that can serve a variety of functions, including as antimicrobials (*Pandith et al., 2019*). Many important human antimicrobials are produced in bacteria *via* this gene cluster, including the antibiotics streptomycin and erythromycin (*Flores-Sanchez & Verpoorte, 2009*). Depending on their structure, polyketide synthases can be classified as Type I, Type II or Type III. Type III polyketide synthases consist of a single enzyme that does not require an acyl-carrier protein and can directly catalyze reactions of acyl-CoA substrates (*Pandith et al., 2019*). While we found Type III polyketide synthase clusters to

be fairly common, we did not see clear evidence that they were linked to Bd inhibition in the genera we examined. For instance, our two *Bacillus* isolates that varied in Bd inhibition had 100% sequence similarity in the seven shared genes within their Type III polyketide synthase cluster. Even for the genera that had some differences in genes present in the Type III polyketide synthase clusters, like in *Microbacterium*, the genes involved seemed unlikely candidates for driving variation in antifungal activity, as there was no clear association between genetic differences and Bd inhibition in our dataset.

The final biosynthetic gene cluster that we focused on were the siderophores. Iron is a necessary nutrient; however, soluble forms of iron are scarce in the environment (*Andrews, Robinson & Rodríguez-Quiñones, 2003*). Siderophores help bacteria sequester iron from the environment, and they may, therefore, limit iron availability to fungi and inhibit fungal growth (*Scher & Baker, 1982*). For example, siderophores can inhibit plant fungal pathogens (*Scher & Baker, 1982*). In our dataset, the nucleotide sequences of the 6 shared siderophore genes were 100% identical in two of the *Bacillus* isolates that differed in their ability to inhibit Bd growth, and thus are not likely critical for Bd inhibition. As we had only two *Pedobacter* isolates, we could not examine whether the variation in their shared siderophore genes is likely to contribute to variation in ability to inhibit Bd growth.

We also considered genes for several individual compounds in our analysis, namely chitinase and violacein. Chitin is a major component of the fungal cell wall and is a polysaccharide composed of N-acetylglucosamine with beta 1,4 linkages (*Lenardon, Munro & Gow, 2010*). Chitinases are enzymes that can cleave the beta 1,4 glycosidic linkages, which results in degradation of fungal cell walls (*Le & Yang, 2019*). The two *Agrobacterium* isolates in our dataset varied in Bd growth inhibition. Both of them contained a predicted chitinase, but it differed in four amino acids, which could alter the structure, and thus functionality, of this predicted chitinase enzyme. In addition, the more inhibitory of the two isolates also contained a second predicted chitinase. The only inhibitory isolate of the five *Sphingomonas* isolates also contained a predicted chitinase. These results suggest that chitinase may be a compound worth investigating more in terms of Bd growth inhibition by symbiotic bacteria.

Violacein is a compound produced by *Janthinobacterium lividum* that can inhibit growth of both Bd and *B. salamandrivorans* (*Woodhams et al., 2018*). Three of our *Janthinobacterium* isolates contained all of the necessary genes, vioA-E, for violacein production. Interestingly, the fourth *Janthinobacterium* isolate contained only vioA, and thus is unlikely to be producing violacein, even though it still had some ability to inhibit Bd. Two of the isolates that contained all the violacein genes were strongly inhibitory, while the third was much less inhibitory. We found that many amino acid substitutions were only present in the less inhibitory isolate. While there were amino acid substitutions in both the vioA and vioE protein sequences, both sequences contained the amino acids that are critical for their functionality (*Füller et al., 2016*; *Hirano et al., 2008*). This indicates that the differences seen in the vioA and vioE genes most likely do not contribute to variation in the ability to inhibit Bd growth. Less is known about the structure of the proteins vioB, vioC and vioD. We hypothesize that these differences within the vioB, vioC and vioD genes could explain the variation in Bd inhibition.

It is interesting to note the variation in Bd inhibition, and genomic variation, that occurs within bacterial genera isolated from the same population of amphibians. Both *Agrobacterium* isolates were isolated from the same population of spring peepers on the same day; however, one showed a stronger ability to inhibit Bd growth, and concordant changes in putative chitinase genes. This was also seen in the five *Sphingomonas* isolates, which were also all isolated from the same population of spring peepers. Similarly, three of the *Janthinobacterium* isolates were cultured from the same population of American toads, and showed variation in both their ability to inhibit Bd and the genes necessary to produce violacein. This suggests that genus-level taxonomy is not adequate for predicting bacterial function against Bd in these symbiotic communities. Even closely related bacterial taxa can sometimes have key genetic differences that result in different functional roles in communities (*Becker et al., 2015b*).

## CONCLUSION

Our results stem from computational predictions, and therefore can only serve to generate hypotheses about the function and importance of these specific genes and their links to *in vitro* Bd inhibition of the isolates. Based on our findings, we would recommend both chitinase and violacein genes as candidates for future studies, but there are likely many other compounds and pathways important for anti-fungal activity in these skin bacterial communities. For example, a metagenomic study found several gene classes that were more abundant in bacterial communities from frogs that lived in an area infected with Bd than frogs who had not been exposed to Bd. These gene classes were involved in biosynthesis of secondary metabolites, cellular communication, and membrane transport (*Rebollar et al., 2018*). They also found genes that are associated with known Bd inhibitory compounds such as prodigiosin, 2,4-diacetylphloroglucinol and indole-3-carboxaldehyde. Another study has also highlighted the potential role of prodigiosin, produced by *Serratia marcescens*, in mediating bacterial-bacterial interactions in the amphibian skin microbiome (*Madison et al., 2019*). These studies, and our findings, suggest that combining genomic and metagenomic studies with experimental approaches may be the best way to begin to unravel the complex interactions and functions in host-associated microbial communities.

### Funding
This work was supported by the Morris Animal Foundation, Fralin Life Science Institute at Virginia Polytechnic Institute and State University, and the National Science Foundation (DEB-1136640). The funders had no role in study design, data collection and analysis, decision to publish, or preparation of the manuscript.

### Grant Disclosures
The following grant information was disclosed by the authors:

Morris Animal Foundation, Fralin Life Science Institute at Virginia Polytechnic Institute and State University.
National Science Foundation: DEB-1136640.

## Competing Interests

The authors declare there are no competing interests.

## Author Contributions

- Noah Wax conceived and designed the experiments, performed the experiments, analyzed the data, prepared figures and/or tables, authored or reviewed drafts of the article, and approved the final draft.
- Jenifer B. Walke conceived and designed the experiments, performed the experiments, analyzed the data, authored or reviewed drafts of the article, and approved the final draft.
- David C. Haak conceived and designed the experiments, analyzed the data, authored or reviewed drafts of the article, contributions equal to Lisa Belden, and approved the final draft.
- Lisa K. Belden conceived and designed the experiments, analyzed the data, authored or reviewed drafts of the article, contributions equal to David Haak, and approved the final draft.

## Animal Ethics

The following information was supplied relating to ethical approvals (*i.e.*, approving body and any reference numbers):

Virginia Tech Institutional Animal Care and Use Committee

## Data Availability

Raw data are publicly available via NCBI under Project PRJNA922551. NCBI requires assemblies to be length trimmed, untrimmed assemblies, and all processing scripts (BASH, R, and Python) used for this research are available from the Virginia Tech Data Repository via this link: https://doi.org/10.7294/16832692.v2.

## Supplemental Information

Supplemental information for this article can be found online at http://dx.doi.org/10.7717/peerj.15714#supplemental-information.

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
