# Peer review of "Comparative genomics of bacteria from amphibian skin associated with inhibition of an amphibian fungal pathogen, Batrachochytrium dendrobatidis"

_PeerJ, doi:10.7717/peerj.15714_

## Round 0.1 · original submission · Major Revisions

Dear Dr. Wax and colleagues:

Thanks for submitting your manuscript to PeerJ. I have now received two independent reviews of your work, and as you will see, the reviewers raised some concerns about the research. Despite this, the reviewers are optimistic about your work and the potential impact it will have on research studying amphibian skin microbiota impacts on fungal pathogens. Thus, I encourage you to revise your manuscript, accordingly, taking into account all of the concerns raised by both reviewers.

While the concerns of the reviews are relatively minor, this is a major revision to ensure that the original reviewers have a chance to evaluate your responses to their concerns.

Importantly, please revise your writing and topic organization for clarity. The presentation is a bit disjointed. Also, please ensure that all sequences are submitted to a public database.

There are many minor suggestions to improve the manuscript (typos, nuances, etc.).

I agree with many of the concerns of the reviewers, and thus feel that their suggestions should be adequately addressed before moving forward.

I look forward to seeing your revision, and thanks again for submitting your work to PeerJ.

Good luck with your revision,

-joe

Reviewer 1 ·

Basic reporting

Overall, the authors did a great job! This paper adds to the literature on our understanding of how bacterial-fungal pathogen interactions occur and provides some really interesting insight into potential causes of variability in inhibition abilities seen across isolates. I have only minor comments.

It would be nice to see the authors submit sequences to public repositories such as Genbank where others can access and would likely result in more citations for this paper.

Experimental design

The authors did a nice job presenting the methods, and I really appreciated the restraint in applying hypothesis testing when it would be inappropriate.

Line 111: Are these values the inhibition scores that you refer to throughout? If so at line 246 you have a value > 100, what does that mean if 100 is complete inhibition?

Validity of the findings

Line 169/Figure 1: You say that there is no clear pattern of Bd inhibition based on phylogenetic relatedness but it looks to me like the clade in the bottom right quadrant is where the majority of Bd inhibitors are?
Line 275: Unless I am missing it, you say that the differences in these genes (7 genes in Type III polyketide synthase clusters) may contribute to variation in ability to inhibit Bd growth but fail to mention how/that they differ and how the isolates in Pedobacter differ in inhibition abilities. Just providing more context for that statement will help readers.
Line 275 – 281: With the first group (Bacillus) I understand how you came to the conclusion that it is unclear if those differences in sequences contribute to the variation in inhibition observed across isolates, however for the rest I do not understand your conclusions, maybe just adding a little more here would be helpful for readers.
Figure 4: Consider not including this as you do a nice job verbally explaining this in the results.
Figure 5 : Instead of having a legend can you incorporate the inhibition scores into this figure somehow? Even just shifting the legend so it lines up would be more helpful.
Figure 6 is not referenced in the text.

Additional comments

Figure 1 legend: I am uncertain but if the “[19]” is a citation you might need to change that back to match the others throughout the text.
Line 272 to 273: I think that the “(no Bd inhibition, -2) and (strong Bd inhibition, 100) either you should drop the ‘and’ or merge the two statements to be in one set of “()”.
Line 305: VioE-V is capitalized here but not elsewhere.
Line 330: you have a numerical citation [44]

Reviewer 2 ·

Basic reporting

This reads as a rough draft of a manuscript that needs more organization, reduction of wordiness, and grammatical editing. This is mostly for the methods and results. For grammar specifically, many sentences lack clear subjects, are run-on, or have tense changes. A lot of extra words such as “the”, “these”, and “for example” could also be deleted (more preference here)- I make suggestions throughout below.

Experimental design

no comment

Validity of the findings

I also think a better presentation and summary of comparison of high and low inhibitory isolates is warranted as the main point of the paper is to compare which genes stand out across gradients of Bd inhibition. Even if this is done is summary format in both results and discussion, it will help the reader understand main findings based on this objective. Along with this presentation a better synthesis and argument of why it's concluded that certain gene clusters are good candidates for Bd inhibition. I believe the data are there they just need to be presented more clearly.

Additional comments

Overall, this is a great study and important information to contribute to the world of amphibian microbiomes and disease. The methods are sound, and results provide candidate genes for further exploration. My main concern is with the basic reporting as mentioned in that section above. See also some line by line comments below:

Introduction
Need a bit more organization as topics jump around a bit and is hard to follow with also some rearrangements o have topics move toward focusing on the point of this paper at the end. Right now appears to be an assortment of facts without a clear direction towards the point of the paper. Also there is a lot of “intro-type” material that could be moved here that appear to be out of place in other sections that would help the reader be introduced to which gene families were the focus of this study and why.
38 I would stress that the word microbiome at least according to stricter definitions in the literature includes the abiotic factors along with the biotic factors that comprise an ecosystem- perhaps can include this or use the term microbial community?
42 Awkward placement of this sentence since it does not address any immune defenses perhaps a rearrangement or removal of “for example”?
48 new paragraph perhaps and replacement of word “these” with better description of what exact bacterial communities you are referring to since this sentence is a bit removed of the first introduction of microbial communities.
71 – perhaps list some of these applications that are highlighted in the cited literature?
93- suggest remove “the” and other “the” in similar format throughout.
94-94 : need to double check official scientific names I believe it’s American bullfrogs, and for newts it’s either the species only of “Eastern newt” or the subspecies designation “Red-spotted newt” I think either are appropriate just need with the species of subspecies designation in scientific name to match. I am using the SSAR database.
106 suggest remove the
110 suggest remove in the assay
116 It would be good to have a table of each isolate, it's taxonomy (at least to the lowest available rank) and the Bd inhibition scores for each. I think there's many ways to incorporate these data into a chart/figure but how it’s presented currently is a bit hard to interpret and no list of the actual inhibition scores is easily accessible and only genus is listed in any figures
133,134 suggest delete the, all
178-191 these sentences could be combined and shortened to be more clear and concise
170 no clear phylogenetic or taxonomic pattern? Perhaps delete this or move as this is results.
176 are these subsets based on genera that are already identified to have these genes or is this a subset of any of those you were able to look at the genus level?
179-181- this appears to be repeated info that belong above before line 175?
186 “PIRATE was run on further divisions of class…”
188 I think this sentence belongs in front of the sentence above and can be combined, otherwise it read as if you did a second class analysis- perhaps it is what you did and I’m misinterpreting
194-197 I understand that some of your methods were dependent on results of the first part of your methods but a lot of this seems to be more appropriate in results and is a nice summary I feel is lacking in results
210-219 Most of this detail could be moved to results and I think would help focus the methods. I think only which isolates were chose and what analyses were done would be sufficient without getting into the details of what was found.
239-247 again most of this detail could be saved for results with only general methods staying- for example: “ Annotated genomes from individual isolates were then analyzed for genes that could potentially explain variation in Bd growth inhibition assays. We looked for predicted chitinase genes and those coding for violacein. For chitinase we did XXX. For genes that code for violacein we did YYY”

Results

279-281- Why not? Perhaps meant for discussion?
283 “perhaps emphasize here how different the inhibition scores are perhaps “with disparate inhibition….” Also good to mention here at the beginning that two were in Agrobacterium and one was in Sphingomonas.
286 this was hard to read and I initially thought you referring to more isolates perhaps delete “less and more inhibitory” and replace with “these two”
292-307 there's good stuff in here but he organization makes it hard to follow to understand what differences really stand out between those isolates of varying inhibition score. I believe also tense should stay in past tense
309 this would be a great sentence in the results, minus the “surprising’ comment
Discussion
For each paragraph focusing on different gene clusters- there is good intro information that I believe better belongs in the introduction as this the first time the reader is exposed to why you may have chosen these gene clusters and their importance.

335- can you delve a bit more deeply into why you believe this? I know no statistics were done so perhaps just a bit more details and why your initial survey suggest this.
344-346- same as above a bit more synthesis here
371-373 I think this sentence need to be re-written to be more clear
371-379- watch common names for consistency in capitalization, and any new sp needs a scientific name
379-382- a good synthesis here
394-396 – also I think belongs in intro

---

## Round 0.2 · accepted · Accept

Dear Dr. Wax and colleagues:

Thanks for revising your manuscript based on the concerns raised by the reviewers. I now believe that your manuscript is suitable for publication. Congratulations! I look forward to seeing this work in print, and I anticipate it being an important resource for groups studying amphibian skin microbiota impacts on fungal pathogens. Thanks again for choosing PeerJ to publish such important work.

Best,

-joe